# Scientometric Analysis and Systematic Review of Multi-Material Additive Manufacturing of Polymers

**DOI:** 10.3390/polym13121957

**Published:** 2021-06-12

**Authors:** Yufan Zheng, Wenkang Zhang, David Moises Baca Lopez, Rafiq Ahmad

**Affiliations:** Laboratory of Intelligent Manufacturing, Design and Automation (LIMDA), Department of Mechanical Engineering, University of Alberta, Edmonton, AB T6G 1H9, Canada; yufan6@ualberta.ca (Y.Z.); wenkang@ualberta.ca (W.Z.); bacalope@ualberta.ca (D.M.B.L.)

**Keywords:** additive manufacturing, multi-material, polymers, review, scientometric analysis

## Abstract

Multi-material additive manufacturing of polymers has experienced a remarkable increase in interest over the last 20 years. This technology can rapidly design and directly fabricate three-dimensional (3D) parts with multiple materials without complicating manufacturing processes. This research aims to obtain a comprehensive and in-depth understanding of the current state of research and reveal challenges and opportunities for future research in the area. To achieve the goal, this study conducts a scientometric analysis and a systematic review of the global research published from 2000 to 2021 on multi-material additive manufacturing of polymers. In the scientometric analysis, a total of 2512 journal papers from the Scopus database were analyzed by evaluating the number of publications, literature coupling, keyword co-occurrence, authorship, and countries/regions activities. By doing so, the main research frame, articles, and topics of this research field were quantitatively determined. Subsequently, an in-depth systematic review is proposed to provide insight into recent advances in multi-material additive manufacturing of polymers in the aspect of technologies and applications, respectively. From the scientometric analysis, a heavy bias was found towards studying materials in this field but also a lack of focus on developing technologies. The future trend is proposed by the systematic review and is discussed in the directions of interfacial bonding strength, printing efficiency, and microscale/nanoscale multi-material 3D printing. This study contributes by providing knowledge for practitioners and researchers to understand the state of the art of multi-material additive manufacturing of polymers and expose its research needs, which can serve both academia and industry.

## 1. Introduction

Additive manufacturing (AM), also referred to as three-dimensional (3D) printing, describes a process where 3D objects are produced by combining materials, usually in a layer-by-layer manner, from a 3D model design, as opposed to subtractive manufacturing strategies [1]. This technology has been applied and customized into a wide applications spectrum with recent interests such as automotive, aerospace, medical, dental, building, biological system, electronics, and food supply chains [1,2]. Over traditional manufacturing strategies, AM has demonstrated various advantages, including greatly enhanced design freedom [3], simplified supply chain management [4], maximum material saving, personal customization, and a reduced environmental impact [5].

A wide range of materials have been currently utilized in AM technologies, including metals, ceramics, concrete, and polymers. Metals and alloys have been popular in the aerospace industry to build, repair, and remanufacture various components due to their numerous advantages, particularly their design flexibility, low material wastage, and their lack of scarifying metals’ mechanical properties [6]. Ceramics are usually used for printing biocompatible scaffolds. AM of ceramics has more challenges than AM of polymers or metals due to the former’s high melting points and hence the low tolerance to processing defects of this class of materials [7]. Concrete is the main material employed in the AM of constructions, and AM technologies have opened new building design possibilities [8]. Polymers consist of macromolecules composed of repeating subunits, and they can be classified into thermoplastics and thermosets depending on their thermomechanical properties. Among all AM materials, polymers have been considered as the most common materials in 3D printing because of their diverse material selection and ease of adoption to different AM technologies. Polymer printed parts also have their advantages through their lightweight, corrosion-resistant properties and their achievable mechanical, thermal, electrical, fire-resistance, and biocompatible properties [9]. In addition, by using the new functional polymers with novel biological, mechanical, and chemical properties, new branches of AM also emerged, such as bio-printing and four-dimensional (4D) printing [10]. Polymers for 3D printing are extensively used in the form of photosensitive resin, thermoplastic filaments and powders, and viscous polymer inks [9,11]. The techniques for AM of polymers can be divided into vat photopolymerization [12,13,14], material extrusion [15,16,17], powder bed fusion [18,19,20,21], material jetting [22,23,24], binder jetting [25,26], and sheet lamination [27,28,29]. The working principles of individual AM techniques for polymers differ on the machine technology and material selection. Vat photopolymerization is an AM process that uses radiation (e.g., ultraviolet (UV) light) to selectively scan and polymerize liquid photosensitive resins in a vat to form solid 3D models. The 3D printed part by this technology has higher part accuracy and resolution compared to that fabricated by other AM processes [30,31]. The material extrusion technology is also widely known as Fused Deposition Modeling (FDM). This process uses thin thermoplastics filament, viscous inks, or polymer pellets with a specific size extruded by a nozzle controlled by numerical control (NC) to form the 3D object layer-by-layer. Powder bed fusion utilizes a heat source (e.g., laser or infrared (IR) radiation) to melt and bind powder particles in a powder bed to build 3D models. This process is carried on in an enclosed chamber, which is filled with nitrogen gas to decrease the oxidation and degradation of the material to a minimum. The material jetting printing technique is similar to 2D inkjet printing. In this process, continuous stream or individual drops are ejected from the nozzle and deposited on a platform, subsequently solidifying by photopolymerization and cooling. Binder jetting (also called inkjet-printing) is a powder-based AM technology that uses a print head to deposit the droplet of a liquid binder onto the powder bed and selectively glue them together to form the 3D structure. AM’s last technology for polymers is sheet lamination, which builds a 3D object by stacking and laminating thin sheets of material together. This technology is relatively cheaper than others, so it is usually used for building large parts [9].

Initially, most AM technologies for polymers were developed to use a single material. However, the 3D printed object with a single material cannot meet the rising demand for high complexity and enhanced functional performance. Multi-material additive manufacturing allows different materials to be deposited in various regions or mixed by a percentage ratio in the same region, which can fabricate an object with a wide range of properties and functionalities. This technology can also decrease the production time without extra cost for fabrications with a complex morphology [32]. Recently, various multi-material additive manufacturing technologies for polymers have been developed and implemented in different applications. Therefore, a comprehensive and deep understanding of this technology is required. Whereas existing review publications showcase detailed analyses on certain areas of research, the technologies and applications of multi-material additive manufacturing for polymers have been diverse and with varying degrees of complexity; thus, this study was proposed to provide a full scope of analysis in this research area.

Scientometrics includes quantitative study of science, communication of science, and science policy [33], which reveals the research impact of publications, researchers, journals, and research institutions in a certain field of research and provides a deeper understanding of scientific citations [34]. This study aims to conduct a scientometric review and analysis of the publications relating to the multi-material additive manufacturing of polymers to achieve a comprehensive understanding of the development in this research field over the last two decades (2000 to 2021). Based on the results of scientometrics analysis, an in-depth systematic review is subsequently presented to provide deeper insights into the technologies and applications of multi-material additive manufacturing of polymers.

The rest of the paper is organized as follows. In Section 2, the research methodology is described in detail. Subsequently, the results and findings from the scientometric analysis are discussed in Section 3. In Section 4, a systematic review of the existing literature is proposed. Finally, a conclusion is given in Section 5.

## 2. Research Method

This work’s motivation is to provide an up-to-date review of the research works for multi-material additive manufacturing of polymers and an understanding of the future direction in the field. For this study, a mixed-review method was employed, including steps of data acquisition, scientometric analysis, and systematic review, allowing the research to be analyzed from different perspectives. The flowchart of the methodology is presented in Figure 1. In the first stage, the relevant research results were retrieved from the academic literature database (Scopus) following the meta-analyses guidelines [35]. A scientometric analysis was conducted to measure the targeted research field and map the current knowledge and research topics in a domain based on the retrieved literature. A systematic review was implemented in the last stage, discussing the topics of different technologies and applications for multi-material additive manufacturing of polymers. The topics were carefully selected by following the results of the scientometric analysis. After a manual literature screening, the existing studies were comprehensively reviewed, and the future research directions were also discussed. The subsections explain the research method in detail.

### 2.1. Data Acquisition

Data acquisition of the existing literature was significantly important in this research, especially for the conclusion drawn from the scientometric analysis. For this research, the data acquisition pursued the strategy suggested by [36,37]. The literature collection strategy followed two criteria, including (1) contemporary and relevance: all publications were searched from 2000 to 2021 and the papers were manually literature screened by reviewing the keywords and abstracts carefully; (2) quality assurance: only peer-reviewed papers from journals were included since the journal papers normally undergo careful reviews to eliminate errors and mistakes. 

The database selection was critical for the literature review. In this study, the Scopus database was selected as the literature database due to its extensive coverage of journal publications and knowledge domain compared to other databases such as Web of Science, Google Scholar, or PubMed [34,36].

The existing literature related to multi-material additive manufacturing of polymers in the Scopus was retrieved by searching keywords within the publications’ title/abstract/keywords. Based on the objective of this review, the selected keywords were: (“additive manufacturing*” OR “print*”) AND (“polymer*”) AND (“multi-material*” OR “composite*” OR “dissimilar*”). It should be noted that the wildcard character * was used to capture variations of one keyword. The search period was set to from 2000 to 2021, representing the contemporary development of multi-material additive manufacturing of polymers. A screening process was performed to limit papers published in peer-reviewed English journals. In the end, a filter by subject area was applied to exclude irrelevant fields, such as “Business, Management and Account” and “Social Sciences”. Those remaining papers after the screening process were fed into the scientometric analysis. The initial search yielded 4767 document results, and after the successive screening processes, the number of documents fell to 2512.

### 2.2. Scientometric Analysis

The term “Scientometrics” was first created by Nalimov and Mulchenko in 1969 [38] as “a measurement of science.” The research area of scientometrics has been developed from the second half of the 19th century until today. Over these 100 years, the studies of scientometrics moved from the unconscious to consciousness, from qualitative research to quantitative research, and from external description to comprehensive study disclosing the inherent properties of scientific production [39]. In the recent past, the power of scientometrics was demonstrated in different research areas, such as computer vision in construction [34], construction and demolition waste [40], virtual reality applications for the built environment [36], leak detection and localization [41], smart city [42], sustainability and sustainable development [43], off-site construction [44], public-private partnerships [45], oral health literacy [46], carbon footprints [47], recommendation systems [48], cloud computing [49], and unfrozen soil water [50]. From these literature reviews, the modern scientometric analysis allows us to map knowledge structures, access scientific contribution, find scientific development, and identify emerging trends within a given research field [51]. 

Due to the wide spectrum of research topics related to multi-material additive manufacturing of polymers, it is very challenging to characterize this area’s overall field only by systematic analysis. Although systematic analysis is able to provide an insightful understanding of the research area, it is prone to bias and limited in terms of subjective interpretation [52]. Therefore, in this study, a scientometrics analysis method was proposed to review the previous research results in the area of multi-material additive manufacturing of polymers. Analyses from various perspectives were implemented, including a number of publications analysis, literature coupling analysis, keyword co-occurrence analysis, authorship analysis, and countries/regions activities analysis. In this study, an open-source software, VOSviewer [53], was employed for network modeling and visualization. It is worth noting that the scientific mapping of research communities and themes generated from literature coupling analysis and keyword co-occurrence analysis were both considered in formatting research topics for the subsequent systematic review.

### 2.3. Systematic Review

The identified research clusters can clarify the knowledge domain structure of the multi-material additive manufacturing of polymers; however, the scientometric analysis cannot disclose the in-depth research challenges and inform research needs. Therefore, in this study, a systematic review was conducted as a complementation of the scientometric analysis. Firstly, the authors divided the systematic review into two aspects: technologies and applications. The classification structure of research topics in these two aspects was determined by a consensus-based discussion on the scientometric review analysis. 

## 3. Results and Analysis

### 3.1. Number of Publications Analysis

By applying the literature search method explained in Section 2, 2512 journal articles were published from 2000 to 2021. Figure 2 shows the annual publication number in journals on the topic of multi-material additive manufacturing of polymers. This figure indicates an overall upward trend from 2000 to 2012, and the annual percentage growth rate of the number of publications in this period was +27.5%. Starting from 2012, a burst can be discerned, and the dramatic increase lasted until the year 2020. From 2012 to 2020, the annual percentage growth rate of the number of publications was +184.7%, marking an increase at an incredible rate. Notably, the burst starting from 2012 matched with a milestone of additive manufacturing technology [1,54]. The increasing amount of research for multi-material additive manufacturing of polymers in recent years may be attributed to the increasing availability of mature additive manufacturing technology. In the end, a linear regression was performed based on the data from 2018 to 2020, and the result showed that the number of publications maintained an upward trend and estimated that 677 articles would be published in the year 2021. 

### 3.2. Literature Coupling Analysis

Table 1 summarizes the journals (minimum number of documents of a source is 20) that published articles related to the multi-material additive manufacturing of polymers. The number of publications, number of total citations from these articles, and number of average citations per article were applied to determine the impact of the academic topic in production and research. The majority of academic articles were published in the top journals in the research area of additive manufacturing and materials, including *Additive Manufacturing, ACS Applied Materials, and Interfaces, Composites Part B Engineering, Composites Science and Technology, Scientific Reports, Advanced Materials*, etc. *Additive Manufacturing* (1828 citations, 18.7 average citations per article), *Composites Part B Engineering* (1951 citation, 29.6 average citations per article), *Composites Science and Technology* (1991 citations, 31.1 average citations per article), and *Advanced Materials* (1952 citations, 81.3 average citations per article) can be considered as journals that have a significant influence in this area. Among them, *Additive Manufacturing* includes the most publications (98 articles), *Composites Science and Technology* received the most citations from related articles (1991 citations), and *Advanced Materials* has the highest average citations per article (81.3 citations).

### 3.3. Keyword Co-Occurrence Analysis

Keywords are words that capture the core content of research articles [55] and make the published paper searchable to filter the overwhelming amount of resources available. Keyword network illustrates a knowledge domain to provide penetration into the major research topics and show how these topics are interrelated. In this study, VOSviewer was used for constructing and viewing a keyword network. VOSviewer is a freely available software for building a graphical representation of distance-based bibliometric maps. The distance between two items reflects the strength of the relation between the items [53]. The items in the software can be keywords, authors, organizations, and countries. Generally, a smaller distance represents a strong relation. A label indicates the item, and the label size is directly proportional to the number of publications that include the keyword. The label’s color shows the cluster to which a journal was assigned by the clustering technique of VOSviewer [53].

In this study, the author keyword was used for keywords co-occurrence analysis, and the threshold of keyword occurrences was set to 5. This minimum number of occurrences was determined based primarily on the optimal graphical representation for research clustering by multiple experiments. Therefore, 200 of the 4607 keywords met the threshold. It should be noted that, among these keywords, some words with the same meaning emerged in the map and network, such as ‘3d printing’ and ‘3-d printing’, ‘finite element method’ and ‘finite element analysis,’ and ‘mechanical characterization’ and ‘mechanical properties.’ Furthermore, keywords with a singular form and plural form were required to be combined, like ‘scaffold’ and ‘scaffolds,’ ‘microstructure’ and ‘microstructures,’ and ‘nanotube’ and ‘nanotubes.’ Moreover, the keyword of short-form were merged with its full form, for instance, ‘PLA’ and ‘polylactic acid,’ ‘FEA’ and ‘finite element analysis,’ and ‘CFRP’ and ‘carbon fiber reinforced polymer.’ In the end, to reduce the influence of regular topics, some regular keywords which were not suitable for the specific topic were removed, e.g., ‘3d printing’, ‘additive manufacturing,’ and ‘polymers.’ After the post-processing for manual screening, the keyword network visualization with 155 keyword labels, 906 links, and a total link strength of 1422 is presented in Figure 3.

To obtain precise results, keywords in the quantitative measurement were retrieved. In VOSviewer, the size of each keyword label denotes the weight of occurrence. From Figure 3, it can be observed that ‘mechanical property’ and ‘fused deposition modeling’ appear most frequently. This infers that mechanical property is the most important property for the topic of multi-material additive manufacturing of polymers, and fused deposition modeling is the most popular technique in this area. For a given keyword, the link and total link strength respectively represent the number of linkages of a keyword with other keywords and the total strength of the links of a given keyword with other keywords [53]. 

This study proposed focusing on reviewing different multi-material AM techniques and their key applications. Apparently, the analysis for keywords related to the techniques and applications allowed us to provide some indications. Table 2 lists the keywords related to the techniques. From this table, FDM may be considered as the most dominated technology (highest occurrence: 127; highest link: 65; highest total link strength 176) among other AM technologies in the academic field. This point curiously matches the finding from the survey [10], which illustrated that FDM is popular for multi-material fabrication due to it being relatively affordable and accessible compared to other multi-material AM technologies [56]. In addition, keyword ‘fused deposition modeling’ related articles were most frequently published in 2019, and this can infer that the development of multi-material FDM is still prominent today. The other popular multi-material AM technologies include stereolithography, inkjet printing, selective laser sintering, material extrusion, etc. Selective laser sintering and powder bed fusion are AM techniques for metal or metallic materials; however, these techniques have contributed to hybrid polymer/metal materials [57,58]. Excluding inkjet printing, screen printing, and electrospinning, the average publish year of other techniques are mostly in the range from 2018 to 2020, which means most of them are in the research hotspot. 

Table 3 presents the keywords for multi-material AM applications and their network data. These keywords can be divided into three classes: medical application, electronics, and soft robotics. The application of medical application includes the keywords of ‘tissue engineering’,‘scaffold’, ‘bone tissue engineering’, ‘bone regeneration’, ‘drug delivery’, and ‘biosenso’, which account for 47% of overall occurrences. There are keywords of ‘printed electronics’, ‘flexible electronics’, ‘strain sensor’, ‘electrochemical sensor’, etc., which cover 50% of all keywords occurrences. ‘Soft robotics’ only has 3%. Therefore, it can be inferred that the main applications of multi-material AM techniques include medical and electronic applications.

A new dataset was built by merging the keywords related to applications and technologies to investigate the relationship between applications and techniques. Figure 4 shows the density visualization for keywords of applications and techniques, with 5 clusters. From cluster 1, we could find that stereolithography and inkjet printing are related to applications of soft robotics, actuators, sensor, and dielectrics. Cluster 2 covers various electronics that are related to the technique of screen printing. Cluster 3 indicates that tissue engineering is associated with multi-material AM techniques like fused deposition modeling, selective laser sintering, and material jetting. In Cluster 4, direct link writing, material extrusion, and powder bed fusion are involved, but they are only linked to flexible electronics. In the end, Cluster 5 shows that electrospinning is relevant to applications of stretchable electronics, strain sensors, and temperature sensors. 

### 3.4. Authorship Analysis

The information with respect to the authorships can be acquired from the Scopus database. Therefore, the leading researchers in this research area could be identified, and the collaborations between researchers could be mapped. 

Table 4 lists the authors who published the most publications related to the research field of multi-material additive manufacturing of polymers. The list of authors was acquired from the database, after setting the minimum number of citations of the author as 15. Out of 7346 authors, 15 met the selection criteria. Wang Y. occupied the top position of the most productive researcher. At the same time, in terms of citations, Wang Y. had much higher citations than other authors, which shows that this scholar has the largest influence in this field. 

The collaborations between researchers were visualized by VOSviewer, as shown in Figure 5. The visualization of co-authorship resulted in 4 clusters. Cluster 1 includes Zhang Y., Zhang C., Xu J., Wang Y., Chen Y., Wang Z., and Wang J. From Cluster 2, we can find Zhang H., Zhang Z., Wang L., and Liu Y. In Cluster 3, there are three authors, including Zhang J., Li Y., and Zhang X. Sing R. appears individually in the last cluster, which indicates that there is no collaboration with the other most productive authors in the field. 

### 3.5. Countries/Regions Activities Analysis

The countries/regions actively engaged in the research field of multi-material additive manufacturing of polymers are listed in Table 5. Among them, the United States is dominating the research area in a number of publications and citations. The United States’ citations were found to be 17,544, which is far ahead of other countries/regions—followed by China, which published 378 papers and received 6459 citations. In terms of average citations, Japan and Singapore received 46.5 and 55.4 citations in each publication, which is much higher than other countries/regions.

Figure 6 maps the collaborations between different countries/regions in the research area, which shows 19 nodes and 111 links. The connection lines show the co-authorships between countries/regions. According to the size of the node and thickness of connection lines, the US maintains research links with the rest of the countries/regions.

## 4. Systematic Review of Current Research

### 4.1. Technologies

According to the scientometric analysis, there have already been significant scientific community efforts to fabricate multi-material polymeric 3D parts. This section reviews the related works on various typical 3D printing techniques, including fused deposition modeling (also known as fused filament fabrication), direct ink writing, vat photopolymerization, material jetting, and some innovative hybrid 3D printing platforms. The process mechanism, merit and demerit, and recent research advances of each 3D printing technique are systematically reviewed in the following sections.

#### 4.1.1. Fused Deposition Modeling 

Fused deposition modeling (FDM), also known as fused filament fabrication (FFF), is one of the most employed methods of additive manufacturing technology, in which thermoplastic polymer in the shape of wire (filament) is melted by a liquefier and then extruded with a fine diameter through a nozzle [23,29,59]. The semi-liquid printed material is subsequently added layer-by-layer onto a build platform. In Figure 7a, the scheme of this technique is shown. Compared to other AM methods, the FDM method offers many advantages, including low costs, smooth operations, ease of support material removal, better raw material handling, and the ability to process different thermoplastics [59,60]. 

In the earlier stage of FDM, the printing potential was limited by a small selection of thermoplastic filament materials [61]. Fortunately, with an increasing variety of filament materials offering a wide range of physical, mechanical, and electronic properties, FDM is now highly compatible with a wider range of materials, including acrylonitrile butadiene styrene (ABS) [62,63,64], polycaprolactone (PCL) [65,66,67,68], polylactic acid (PLA) [69,70,71], nylon [72,73,74], polypropylene (PP) [75,76,77], thermoplastic polyurethanes (TPU) [78,79], polyvinyl alcohol (PVA) [80,81], high impact polystyrene (HIPS) [82,83], and composite filaments [84]. Therefore, multi-material 3D printing using FDM has drawn growing interest in recent years. Multi-material FDM can be easily realized by employing multi-material single mixing nozzle or multi-material multiple nozzles (Figure 8a) [85]. It has been demonstrated that the multi-nozzle achieves a better performance in the build time, while the single nozzle shows greater consistency in fabricating high-quality materials [86]. 

However, some challenges associated with the printed parts with the multi-material FDM technology still exist, such as inherent poor surface finish with ridges, limited printing resolution, slow build speed, and low interfacial bonding strength [87]. In particular, the weak bond strength between adjacent extruded filaments and layers is the most critical issue among these limitations because the bonding between dissimilar materials could be a much more intractable problem in multi-material FDM than in single-material FDM because of the differences in physical properties and chemical properties [85]. 

For a better mechanical performance of multi-material FDM printed parts in PLA/TPU material pairs, the significance of interface design was investigated. The results indicate that the macroscopic-based interface geometries designed based on mechanical interlocking systems are more reliable than the simple face-to-face interfaces [89]. To improve the bond strength between dissimilar materials with different melting temperatures, a single-layer temperature-adjusting transition (SLTAT) method was proposed, and this novel technique was found to achieve a 28% increase in the tensile strength compared to unprocessed ones when printing PCL/PLA parts at the 130 °C bonding-layer temperature [88]. As shown in Figure 8b, this process effectively eliminates some defects due to unnecessary heat input by increasing the PCL bonding layer’s nozzle temperature to heat only the top PLA layer above its glass transition temperature while keeping the 3D printing parameters of other layers the same. In addition, the SLTAT method can be easily applied to existing multi-material 3D printers without adding additional heating equipment or post-processing steps (Figure 8b). For the purpose of improving adhesion of dissimilar thermoplastics without the need for chemical compatibilization, a bi-extruder head for FDM systems capable of printing two filaments through a single nozzle was developed [90]. This extruder’s unique feature is its convenient access to the internal channel, which allows the insertion of a static inter-mixer, thus enabling the deposition of mechanically interlocked extrusions consisting of two non-combustible polymers. It has been demonstrated that this innovative FDM extruder could successfully fabricate multi-material 3D parts with enhanced interfacial bonding strength. 

#### 4.1.2. Direct Ink Writing

Direct ink writing (DIW) is an extrusion-based 3D printing technology in which ink is transferred through a nozzle in a regulated pattern under ambient conditions (Figure 7b). After deposition, the ink is immediately cured into a solid object by different post-processes, such as photopolymerization or thermal curing [91,92,93,94]. Similar to FDM, DIW is also a high-efficiency method to deposit multi-material printed parts because it is of low cost and can be simply carried out [95,96,97,98]. More importantly, DIW exhibits tremendous potential because it is highly suitable for printing a large variety of different functional materials through multiple inkjet heads or nozzles to deposit different materials, including metallic particles [99,100,101], ceramic particles [102,103,104], extracellular matrices [105,106,107,108], hydrogels, and elastomers and epoxy thermosets [[109],[110],[111]
[112]]. As a result, DIW with multiple nozzles has become a favored candidate for multi-material 3D printing. 

Nevertheless, this approach often involves a sequential printing process for individual materials, which adds to the build time because of the increased complexity in motion control as well as ink supplies. Tip alignment of each nozzle, which is essential for guaranteeing accurate control of the interface between the extruded lines, may become even more difficult with the increasing number of nozzles [113]. In addition, precise material flow control is required, especially when printing materials with different flow characteristics such as viscosity, surface tension, shear stress, and shear elastic [113,114]. 

Several additions and modifications of the basic DIW settings were performed to address the issues mentioned above. For example, a novel DIW printing method for higher build speed with multi-material 3D printheads was designed and proven to fabricate up to eight materials, each flowing through a network of independently bifurcated channels located within the printhead [115]. To achieve continuous printing with dissimilar materials, a single nozzle with multiple material containers was developed. A microfluidic printhead was designed and fabricated to simultaneously print multiple viscoelastic inks through a single nozzle, as shown in Figure 9a [112]. In this process, the inks are drawn from two opposing syringes and then pushed through two square passageways into a microfluidic interface at the bottom of the nozzle, enabling seamless switching of the two viscoelastic materials. Similarly, a DIW technology integrating a UV exposure system and pneumatic control systems into microfluidic printheads was described (Figure 9b) [116]. The combination of fast curing inks, and the printhead that extrudes and then cures them, allows rapid switching between multiple low-viscosity silicone inks. A dynamic photomask-assisted DIW printing and a two-stage curing hybrid ink were reported (Figure 9c) [112]. In this method, DIW is conducted using a nanocomposite ink containing photocurable resin and thermally curable resin. The dynamic photomask is beneficial for the first stage of light-curing with carefully controlled photocuring time at the pixel level, and the mechanical properties and mechanical property gradient of the printed parts can be further enhanced through the following second-stage thermal curing. This single-nozzle-based photomask-assisted multi-material DIW technique has low cost-effectiveness, strong interface bonding strength, and relatively high resolution for achieving complex configurations and mechanically appropriate gradients in functional applications.

#### 4.1.3. Vat Photopolymerization

Vat photopolymerization is an advanced 3D printing process in which liquid photopolymer in vats are selectively solidified by photo-activated polymerization [114,118]. A series of related techniques based on the vat photopolymerization, such as stereolithography (SLA) and digital light processing (DLP), was developed to process polymeric materials [119,120]. 

In SLA, coherent light sources (usually lasers emitting in the UV-range) are used to induce polymerization and further realize spatially localized cross-linking of the initially liquid polymeric materials (Figure 7c) [121,122]. Compared to other AM methods, SLA exhibits many advantages, including a high printing resolution, high-quality surface finish, and wide material selection. DLP is also a 3D printing technology similar to SLA, where the photopolymer is selectively cross-linked in a layer-by-layer fashion using light to build a free-standing object (Figure 7d) [123,124]. The main difference is that an image is projected onto the photopolymer bath in the DLP process, simultaneously exposing all unmasked areas in the plane. DLP offers the same advantages as SLA and provides a higher printing speed because of the no-scan projection lithography technology. 

Although the multi-material vat photopolymerization technique is possible by using multiple vats containing different materials, it is still technically difficult because of contamination issues and slow transfer speed from one material to another during processing [125,126]. Many innovative AM methods were recently developed to circumvent the technical challenges in multi-material printing based on vat photopolymerization. A multi-material SLA machine equipped with a new rotating vat carousel system, platform assembly, and automatic leveling system was developed [127]. The rotating platform efficiently works to replace the current vat on the printing zone with another vat loaded with a different polymeric material if a material exchange is required, as shown in Figure 10a. However, in this device, interruptive changeover steps and rinse-cleaning between material changeovers add significant additional process time. A novel multi-material SLA method based on aerosol jetting was proposed to achieve direct material supply without utilizing vats [128]. As shown in Figure 10b, this aerosol system with multiple material containers and atomizers increases multi-material printing efficiency by converting liquid materials into small droplets and subsequently curing them locally with a UV laser. Recently, a microfluidic device was integrated with an SLA-based 3D printing system for multi-material fabrication (Figure 10c) [129]. This novel platform requires only a few seconds to perform washing when a material exchange is needed, achieving a speed higher than those of the existing SLA method. Various attempts were also made to achieve multi-material 3D printing based on DLP technology with a shortened cleaning time, such as a vat-free droplet-based DLP with an air jet blower [130], a microfluidic material delivery system (Figure 11) [131], or an active cleaning solution equipped with automated storage and retrieval systems [132]. Although these methods cut down on cleaning time, they still do not eliminate the cleaning process, which still slows down the overall printing speed. Fortunately, a cleaning process-free DLP method for multi-material fabrication based on dynamic fluidic control was recently developed (Figure 12a) [133]. In this method, an integrated fluidic cell, in which multiple liquid photopolymers can be quickly transferred through dynamic fluidic control, can enable switching more than 95% of the material within a few seconds (Figure 12b). Also, because of the elimination of the need for a separate cleaning process, this approach is capable of achieving fast multi-material DLP, even when frequent material changes are needed. 

#### 4.1.4. Material Jetting

Material jetting (MJ) is derived from conventional inkjet printing technology, but unlike mono-layer printing, the building pallets descend sequentially as each successive layer of liquid photopolymer is printed and polymerized, ultimately producing a 3D object [134,135,136,137]. Photopolymer material droplets are selectively dispensed through micro nozzles within the print head, which is composed of a large number of linearly arranged tiny nozzles. They are UV-cured and solidified immediately after the cross-linking reaction by UV light to build a 3D structure, and then the following layer is repeated until the model is successfully established [54,138]. In recent years, MJ printing technology using multiple inks in different colors has become so sophisticated that multi-material MJ printing has been easily accomplished by using multiple printheads in the same way. Each head is equipped with hundreds or even thousands of micro nozzles, allowing for rapid 3D printing with parallel material deposition [139]. 

One limitation with multi-material MJ is that the inkjet nozzle size is so tiny that there is a challenge of printing polymeric materials with viscosities higher than 40 cP [140,141]. To approach this problem, a study on the design and optimization of a high-temperature microheater (more than 600 °C) was carried out, and the results show that this jetting-based method with optimized high-temperature microheater can successfully apply to the manufacture of high viscosity materials because of the reduced viscous dissipation of energy during the printing process [142]. The improved viscosity and good printing behavior can also be realized with small additions of polystyrene as a rheological modifier to the composition of inks, which successfully achieves a 70% increase in viscosity at the expense of 20% of the performance [143]. Another challenge for the multi-material MJ method is the printing resolution of the MJ method limited by current nozzle production techniques because it is highly dependent on the nozzle density of a jetting head. To improve the printing resolution with a given nozzle density, a complete system integrating multi-objective topology optimization into multi-material MJ printing for fabricating complex 3D parts was developed, as shown in Figure 13. In this approach, a topology optimizer dispenses material for a single voxel (volume element) while optimizing physical deformation and high-resolution appearance, and multi-material MJ with a 35 μm resolution was finally demonstrated [144]. 

#### 4.1.5. Hybrid AM Systems

The concept of hybrid 3D printing has enjoyed a steady increase in popularity, as it incorporates multiple 3D printing technologies with various machining tools, such as a robotic arm pick-and-place (PnP) or spray coating technologies, into an integrated manufacturing platform [145,146,147,148]. More importantly, each AM technique has its own strengths and weaknesses, and the range of available materials varies dramatically. For example, DIW 3D printing requires inks with high viscosity and significant shear thinning behavior [149], while inkjet-based 3D printing, such as material jetting, requires low-viscosity inks with tightly controlled physical properties, such as stable surface tension [150,151]. Obviously, different materials have their own unique processing requirement needs while also providing the final 3D objects with multifunctionality. Therefore, hybrid 3D printing, which can take advantage of each 3D printing technique’s unique processing capability to print a wider variety of materials, is more suitable for multi-material 3D objects. 

In recent years, many notable efforts towards AM hybridization were made by incorporating two or more of the printing methods mentioned above into a single platform to create 3D parts. For instance, hybrid AM techniques combining DIW with MJ [152] or DLP [153] were successfully applied to the manufacture of electronic products necessitating the printing of highly viscous conductive materials comprising a high density of metal particles or carbon-based materials. Similarly, a hybrid system combining material jetting and material extrusion AM processes was presented [154]. In this method, a pneumatic material extrusion print head is used to print the outer frame of each layer, while a piezo-pneumatic print head is used to quickly print the fill pattern as well as the support structure (Figure 14a). Compared to the conventional extrusion-based printing technologies, this system cannot only fabricate the non-Newtonian viscous polymeric materials but also enhances the printing speed by between 10 and 20 times. 

Recently, a more advanced methodology for building a multi-material multi-method (m^4^) 3D printer that integrates a wider variety of deposition-based AM technologies (inkjet printing, FDM, DIW, and aerosol jetting) was proposed (Figure 14b) [155]. It was demonstrated by this study that easier manufacturing of 3D devices, including embedded electronics, sensors, soft robotics, and customizable medical devices, could be achieved through this platform. However, the customized m^4^ printer is expensive, and the printing speed is relatively low due to the major use of extrusion-based printing and inkjet printing. To overcome this difficulty, an integration of vat-photopolymerization and material-extrusion methods was implemented. This hybrid platform consists of a top-down DLP printer for high-speed and high-resolution printing of a material matrix with complex geometry and a material-extrusion DIW printer for the printing of functional material, including liquid crystal elastomers (LCEs) and conductive silver inks (Figure 14c) [156]. This work opens a new avenue for printing multifunctional structures and devices through a low-cost, high-resolution, and high-speed integrated printing process, showing great potential for broad applications in areas such as soft robotics, flexible electronics, active metamaterials, and biomedical devices.

### 4.2. Applications

In recent years, multi-material polymer-based products have provided substantial benefits to the global industry and research community, and as a result, novel investigations and technological advancements focused on increasing levels of multifunctionality in various applications. The possibility of fabricating customized multiple materials structures using 3D printing technology enabled specific material selection and enhanced different properties when compared to single homogeneous structures. The following review of recent multi-material polymers with AM applications in the engineering, healthcare, and electronics sectors is presented based on the scientometric analysis. 

#### 4.2.1. Engineering

Many high-tech sectors of engineering benefited substantially from the advancements of additive manufacturing. Among these, the aerospace sector identified the importance of developing and applying multi-material components, since it contributes to lightweight designs and tooling testing launched into space. The extend of these materials include ceramics, metal powders, reinforced composites, and polymer materials [86,157,158]. Also, customization and material performance are essential for the production of multi-material parts, since these characteristics represent a significant role in any aircraft or space mission. Therefore, designing various components such as the heat shield of a space shuttle with multiple-material components using carbon fiber reinforced polymers (CFRP) increases the flexibility and reliability of parts made by AM [56]. Moreover, polymer composites such as CarbonMide, PA, and ULTEM [159] are used in specific locations where relatively low temperatures are identified, such as control panels, fan ducts, conformal cooling channels, and acoustic liner of engines. Besides, sandwich structures polymer composites [157,158,160] fabricated with various reinforcements such as carbon fibers (CF) and thermoplastic polymers can benefit unmanned aerial vehicles (UAV) applications. Figure 15a,b show the clamp of a quadcopter drone fabricated with CFRP layers for improvements in mechanical strength, lightweight, and stiffness [161,162]. Also, AM parts with CF filled with other materials such as resin composites like polyetherketoneketone (PEKK) are being used in aircraft frames and interiors such as air ducts for cabin ventilation in aerospace applications [163]. Reference [164] presents a proof of concept of the applicability of a cost-efficient, automated CF additive manufacturing for morphing aerospace structures (as shown in Figure 15c), which shows great potential in improving flight performance. 

On the other hand, the automotive sector took advantage of AM techniques from rapid prototyping to the production stage using multi-material designs in numerous applications, including engine blocks with built-in temperature sensors [165], custom seats, reinforced car bumpers, multicolored tail-lights, and interior/exterior trims using high-performance materials like ULTEM, the brand name of polyetherimide (PEI), and polyether ether ketone (PEEK) [166]. Also, the supply chain demand for just spare parts in time [167] gained speed for best practices in the automotive industry, and multi-material AM technology proved its benefits by producing customized ergonomic tooling and tough composites for panels and lighter fixtures [168]. Other important applications using multi-material additive manufacturing include designing and fabricating heterogeneous multi-material objects such as flip flops and slippers with desired elastic deformation behavior by varying each material’s internal microstructure [169].

#### 4.2.2. Healthcare

Recent research developments in multi-material AM technology have brought the field of medicine successful progress in bio-inspired fabrications [170], in developing tissue engineering structures for delicate human parts [2,171], and applying biodegradable polymers for cell encapsulation and drug delivery systems [172]. 

Among the thermoplastic polymers used for drug carrier and drug dosage are PVA (polyvinyl alcohol) [173], PVP (polyvinylpyrrolidone), PAA (polyacrylic acid) [174], and PCL (polycaprolactone) with biodegradable stents to prevent blood clots [175,176]. Also, with the use of multi-material parts, different regions of a part can be controlled independently to vary their biological, electrical, and mechanical properties [56]. 

Biocompatible materials are critical for the fabrication of scaffold-based tissue engineering. These biocompatible materials are diverse regarding their mechanical, chemical, and biological properties. Scaffolds for bone regeneration have different mechanical property requirements compared to connective tissues. Regarding bone tissue engineering, implants benefit from the mechanical behavior of different materials such as ceramics and composites to create multiple gradient cellular structures with different cell-based printing techniques for AM technology [85,177]. Composite ceramic and polymer materials combine tunable macro/microporosity and osteoinductive properties of ceramic materials with biodegradable polymers’ chemical/physical properties [178]. Some common composite ceramics and polymers scaffolds fabricated using AM for bone tissue engineering are summarized in Table 6.

Another significant medical application is enabling multi-material 4D printing parts [179] and tailorable shape memory polymers [32,180]. New developments in smart structures for actuation with medical deformable soft robot designs and bio-inspired structures [181,182,183] accelerated the growth in rehabilitation, surgical, and diagnosis in patients. Prosthetics robots were designed and fabricated using multi-material AM processes for individuals with missing limbs or extremities [184,185]. 

Also, in the dentistry field, ongoing research enables printing dentures, custom trays, and aligners directly with additive manufacturing using multiple soft and hard polymers with ceramics [186,187,188].

#### 4.2.3. Electronics

The use of multi-materials is essential for electronics, since it provides for dissimilar material properties to be integrated into a unique three-dimensional circuit design and therefore produce direct manufacturing 3D printed functional embedded components with miniaturized architecture within a reduced footprint. Smart sensor integration [198] and microelectromechanical systems (MEMS) [199] are among the applications being fabricated using multi-material additive manufacturing technologies. Pressure sensors have been created using multi-material extrusion-based systems with different materials, including photopolymers as the base material and ionic liquids for the intermediate layers [200]. Recently, a deformable soft robot actuator with embedded sensors was produced entirely with multi-material 3D printing and incorporated into an end effector gripping system of a robotic arm for assistance into human-occupied procedures [201]. Another study demonstrated the importance of producing fine circuitry with complex and flexible designs by using nanocomposites as conductive materials fabricated with extrusion 3D printed techniques [202]. The FDM process can be used to create embedded sensors with low variability in electrical resistance properties [198,203] and with the increased piezoelectric response using continuous nanocomposite filament [204].

## 5. Discussion and Future Trends

### 5.1. Overview

This study applied a mixed review method that incorporates scientometric analysis and systematic review to explore the current state of research on multi-material additive manufacturing of polymers. The contribution of this study can be considered as extending previous review works in this area by complementing subjective studies with a strong quantitative description and evaluation by science network mapping tools.

As illustrated in Figure 2, the literature on multi-material additive manufacturing of polymers started to increase dramatically from 2012. This trend not only matches the milestone of additive manufacturing technology, but also confirms the growing interest in research in the field of multi-material additive manufacturing. Publications are highly dispersed between 22 different journals (as shown in Table 1). Although journal publications are equally dispersed, *Additive Manufacturing* has the highest number of publications among the total publications (5.02%). *Additive Manufacturing* is considered as the world-leading journal in the area of additive manufacturing, which covers a wide scope, including technologies, processes, methods, materials, systems, and applications. Except for *Additive Manufacturing* and *Rapid Prototyping Journal*, Table 1 suggests that most of the journals focus on materials. Therefore, researchers working on technologies or processes of multi-materials additive manufacturing may encounter issues when deciding where to publish their works. 

This study considered the relationships between keywords in publications. From Figure 3, it can be found that researchers devoted the most efforts into the investigations of the mechanical properties for multi-material additive manufacturing of polymers. However, thermal, electrical, fire-resistance, and biocompatible properties which are significant for new functional polymers in the applications of electronics, bio-printing, and 4D printing attracted much less interests in this field. Another hot spot from the keywords network is fused deposition modeling, which is considered as the most employed approach for multi-material additive manufacturing because of its increasing variety of materials and achievable process. However, the issues associated with the multi-material FDM technology have not been addressed, especially the weak bond strength between different materials, which needs more attention from academia and industry.

The scientometric analysis in this study was able to comprehensively evaluate the existing research results; however, it is limited to quantitively deriving the potential research gaps and future trends. Therefore, the future directions proposed by the systematic review are based on the authors’ knowledge, and they are summarized in the following section. 

### 5.2. Future Trend

Many various 3D printing processes for the fabrication of multiple polymeric materials have been reviewed in this manuscript. It has been indicated that the FDM and DIW methods, which are both based on the material extrusion technique, can be simply extended to multi-material 3D printing by increasing the number of printing nozzles. Similarly, MJ printing technology is a common method for multi-material printing through the use of multiple printheads with inks of disparate materials. PolyJet (Stratasys Ltd., Eden Prairie, MN, USA) is perhaps the most commonly employed commercial multi-material MJ process. It should also be mentioned that the vat photopolymerization process, which typically includes SLA and DLP, is generally not well-suited for multi-material 3D printing because the use of multiple materials in vat photopolymerization, which would result in cross-contamination between different material systems, is challenging. Nevertheless, the technology’s advantages of a high print resolution, high surface finish quality, and wide range of material options still continue to attract much academic attention, which inspired many innovative processes adapted to achieve multi-material printing. Despite the remarkable advances in multi-material 3D printing in the past few years, the potential has not been fully explored yet.

#### 5.2.1. Microscale/Nanoscale Multi-Material 3D Printing

Micro/nano multi-material 3D printing methods have drawn a great deal of concern, as they have implications for many applications such as MEMS/NEMS and nanofabrication. Although most commercial multi-material 3D printers can fabricate macro-scale parts, there are many practical applications where 3D printed parts at different scales are in demand. As mentioned earlier, the hybrid 3D printing platform, which can balance resolution and printing efficiency at the micro-nanometer scale, seems to be a promising printing technique for 3D printing of parts from the micro to nanoscale of different materials [205,206]. 

#### 5.2.2. Printing Efficiency

Printing efficiency is another potential area of research in the field of multi-material AM. A trade-off between printing efficiency (e.g., scanning speed) and part quality (e.g., print resolution) is consistently required. Higher energy power or faster scanning speeds can be employed to improve printing efficiency, but this approach may sacrifice printing accuracy. One solution is to optimize printing parameters through numerical simulation [207], in-situ monitoring [208], or artificial intelligence [209,210] to circumvent this difficulty. Moreover, complex, time-consuming post-processes add to the overall printing time. It is also difficult to scale up the production or print platform, as there are issues with post-processing, like heat treatment processes and removal of support material. Therefore, effective post-processing methods need to be continuously developed and improved. 

#### 5.2.3. Interfacial Bonding Strength

One of the challenges in multi-material 3D printing is to engineer the interfaces between the parts of multiple materials to realize the appropriate degree of interconnection. Although various multi-material techniques have demonstrated great progress and vast potential for future development, the weak bond strength between adjacent different printed layers of different materials is still a difficult problem because of the formation of defects caused by differences in the physical properties of materials and chemical properties, which would finally affect overall mechanical performance of the printed parts. Also, multi-material 3D printing technology can cause anisotropy in the printed part, and the temperature gradient of the material due to layer-by-layer fabrication can also reduce the mechanical properties of each layer. The approaches to improve the mechanical strength of the multi-material 3D parts can be divided into two major groups: processing parameter optimization [211,212] and extra external energy input [213,214]. In addition to technical breakthroughs, fundamental scientific understanding on the inter-layer cohesion mechanisms between dissimilar printed materials is needed to push the boundary of multi-material additive manufacturing.

#### 5.2.4. 4D Multi-Material Printing

Notably, 4D multi-material printing is a novel and fascinating method for producing parts that can adopt new functionalities or shapes after fabrication. Nevertheless, some similar difficulties including low printing efficiency and evolution, a limited range of materials, and insufficient mechanical performance still exist, because 4D multi-material printing was principally developed based on typical 3D printing methods such as DLP, DIW, and FDM. New scientific progress may contribute to the activation or movement of intelligent structures according to a predefined program. Similarly, gradient functional materials can not only control the microstructural properties of 4D printed structures by rationally tuning the density and orientation of the printed material layer-by-layer to create more complex geometric transformations but also enhance the interface bonding strength of different smart compositions [215]. In conclusion, the advancement of multi-material 4D printing is a determinant factor in driving the development of the smart materials domain.

## 6. Conclusions

Multi-material AM of polymeric materials has started to transform certain key aspects of the manufacturing industry for the fabrication of 3D parts and has attracted increasing attention from researchers and practitioners. A scientometric study was proposed to explore the status and global trends of various multi-material 3D printing technologies for processing polymers and their applications. While a number of literature reviews related to the field have been extensively conducted, the scientometric study was presented as a whole for the first time, with 2512 journal papers from the Scopus database investigated through the “scientific mapping” approach. The number of publications, literature coupling, keyword co-occurrence, authorship, and countries/regions activities were analyzed, and in this way, the principal research frames, articles, and relevant research topics were consequently identified. From the scientometric analysis, a heavy bias was found towards studying materials in this field but lacking focusing on developing technologies. Researchers have devoted the most efforts into investigating the mechanical properties on multi-material additive manufacturing. However, thermal, electrical, fire-resistance, and biocompatible properties which are significant for new functional polymers in the applications of electronics, bio-printing, and 4D printing attracted much less interest in this field. Thus, this paper provides an in-depth systematic review of the latest advances in multi-material AM of polymers in terms of technology and applications, respectively. Certainly, various multi-material AM techniques, including FDM, DIW, SLA, DLP, and MJ, provide an outstanding tool for the rapid fabrication of polymer parts with geometric complexity and material diversity. Also, it is driving the development of multi-functional 3D parts, probably leading to revolutionary solutions in a variety of fields, including biomedical engineering, medical devices, and electronics. Although there are many significant achievements of multi-material AM technology, some outstanding challenges still remain, including the printing efficiency, interfacial bonding strength between different materials, cross-contamination, scalability, and applications that are mentioned above. Multidisciplinary research and development will be essential to conquer these challenges, and advances in multi-material AM and its inventive applications in novel fields will expedite scientific discoveries and technological innovations in many domains.

Some limitations exist in the proposed methodology. First, the publications were retrieved from a single database because of the requirement for a consistent data format from VOSViewer. It can be improved by integrating various databases from Scopus, Google Scholar, Web of Science to obtain more comprehensive data. In addition, the scientometric analysis in this study is unable to offer or visualize the future trend directly, while it is summarized by the authors’ knowledge, which may lead to it not being comprehensive. In future work, the publications can be divided into different groups, and results of scientometric analysis in each group will be submitted to the corresponding expert, where experts in different directions can further put forward the research gaps and findings to increase the comprehensiveness of suggestions.

## Figures and Tables

**Figure 1 polymers-13-01957-f001:**
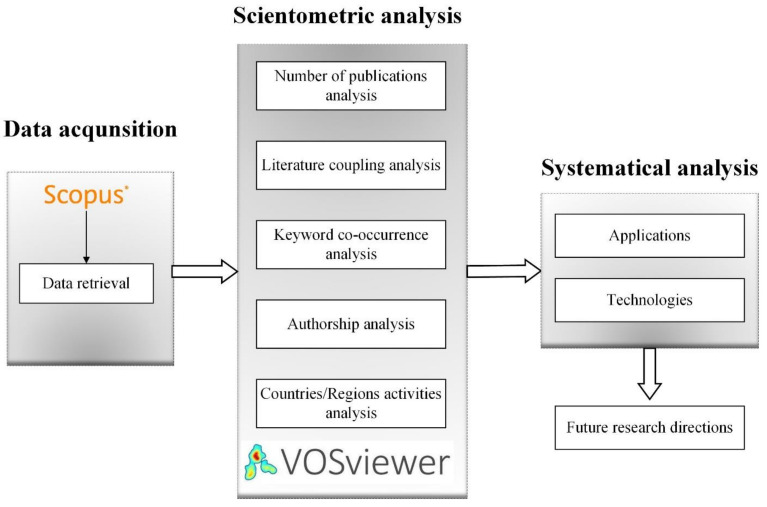
Overall research methodology.

**Figure 2 polymers-13-01957-f002:**
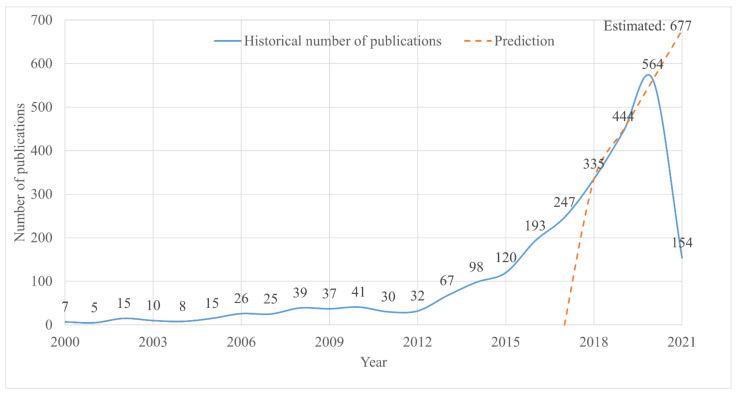
Historical number of papers published each year from 2000 to 2020; the prediction for 2021.

**Figure 3 polymers-13-01957-f003:**
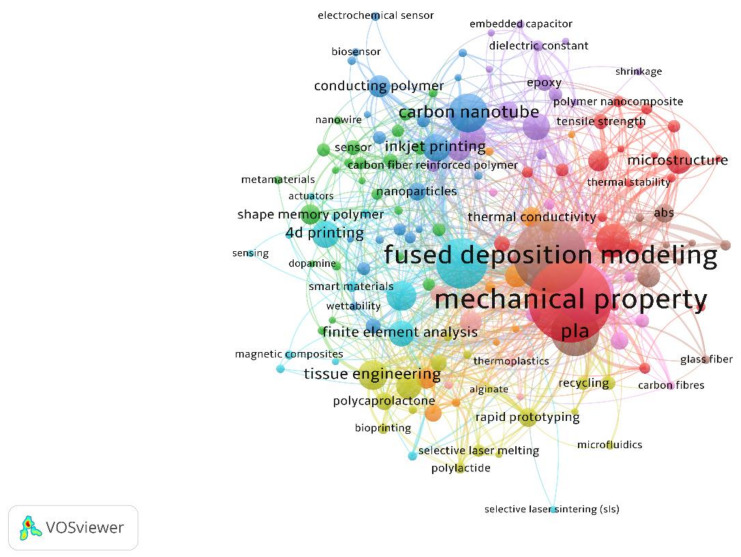
Network Visualization for 155 keywords.

**Figure 4 polymers-13-01957-f004:**
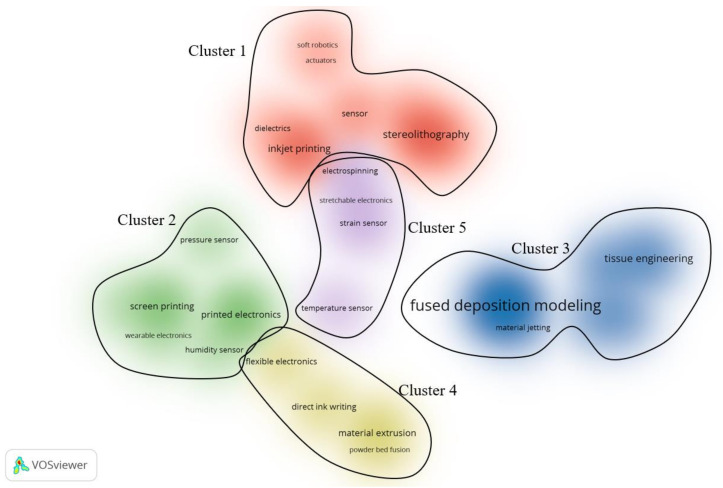
Density visualization for keywords related to technologies and applications.

**Figure 5 polymers-13-01957-f005:**
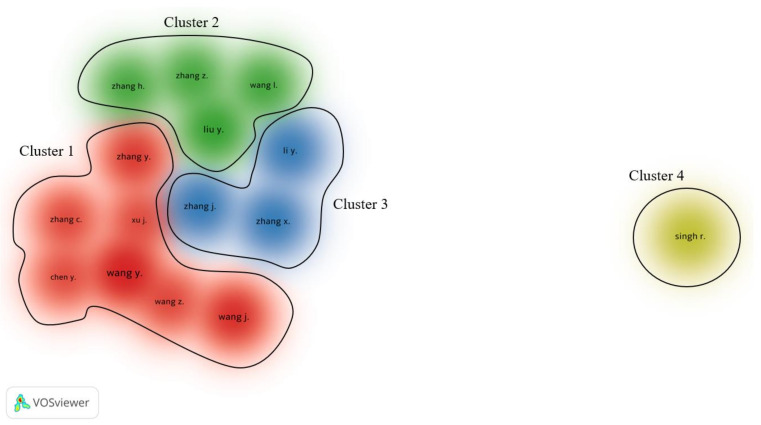
Visualization for co-authorships.

**Figure 6 polymers-13-01957-f006:**
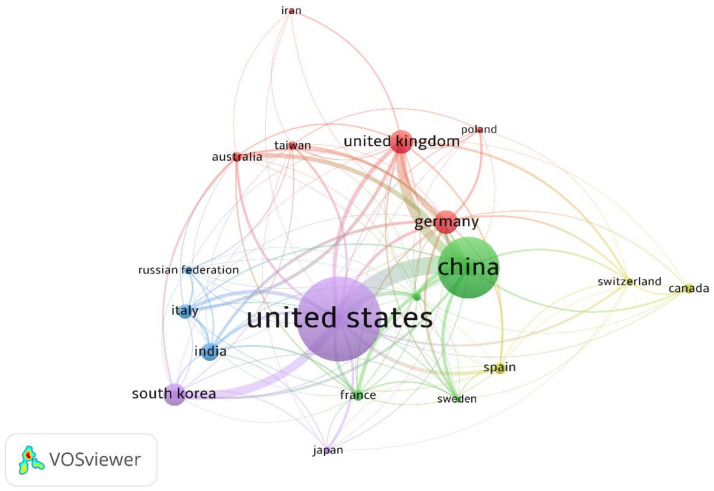
Visualization for collaborations between countries/regions.

**Figure 7 polymers-13-01957-f007:**
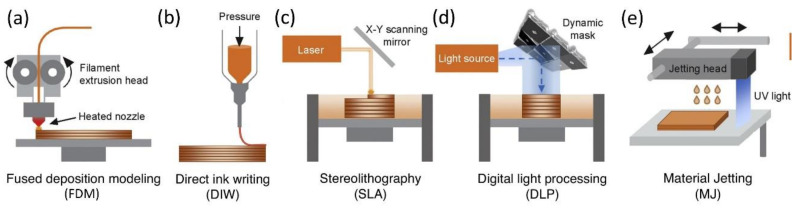
Schematics of five various additive manufacturing methods: (**a**) fused deposition modeling (FDM), (**b**) direct ink writing (DIW), (**c**) stereolithography (SLA), (**d**) digital light processing (DLP), and (**e**) material jetting (MJ). Schematic images were adapted from Ref. [10].

**Figure 8 polymers-13-01957-f008:**
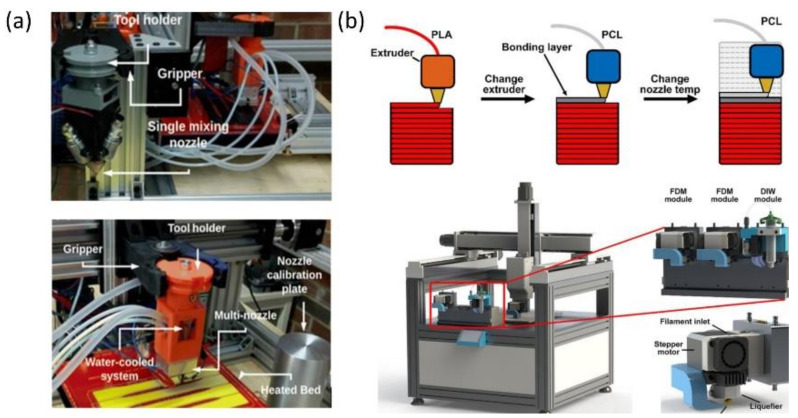
Recent advances in the multi-material FDM technique: (**a**) the setup developed for single mixing nozzle and multiple nozzles for the multi-material FDM system (images were adapted from Ref. [86]) and (**b**) schematics of the principle of the SLTAT method and the corresponding 3D printing platform with a multi-material 3D printer, AMTC system and FDM print-head module (images are adapted from Ref. [88]).

**Figure 9 polymers-13-01957-f009:**
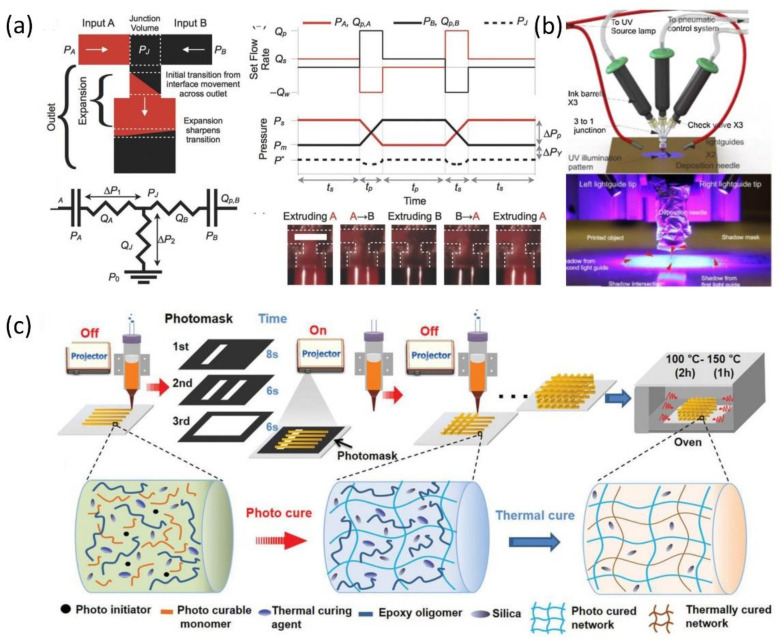
Recent advances in multi-material DIW technique: (**a**) the microfluidic printhead designed to simultaneously print multiple viscoelastic inks through a single nozzle (images were adapted from Ref. [117]), (**b**) the DIW printer integrating a UV exposure system and pneumatic control systems into microfluidic printheads (images were adapted from Ref. [116]) and (**c**) the dynamic photomask-assisted DIW printing technique with an interpenetrated polymer network (images were adapted from Ref. [112]).

**Figure 10 polymers-13-01957-f010:**
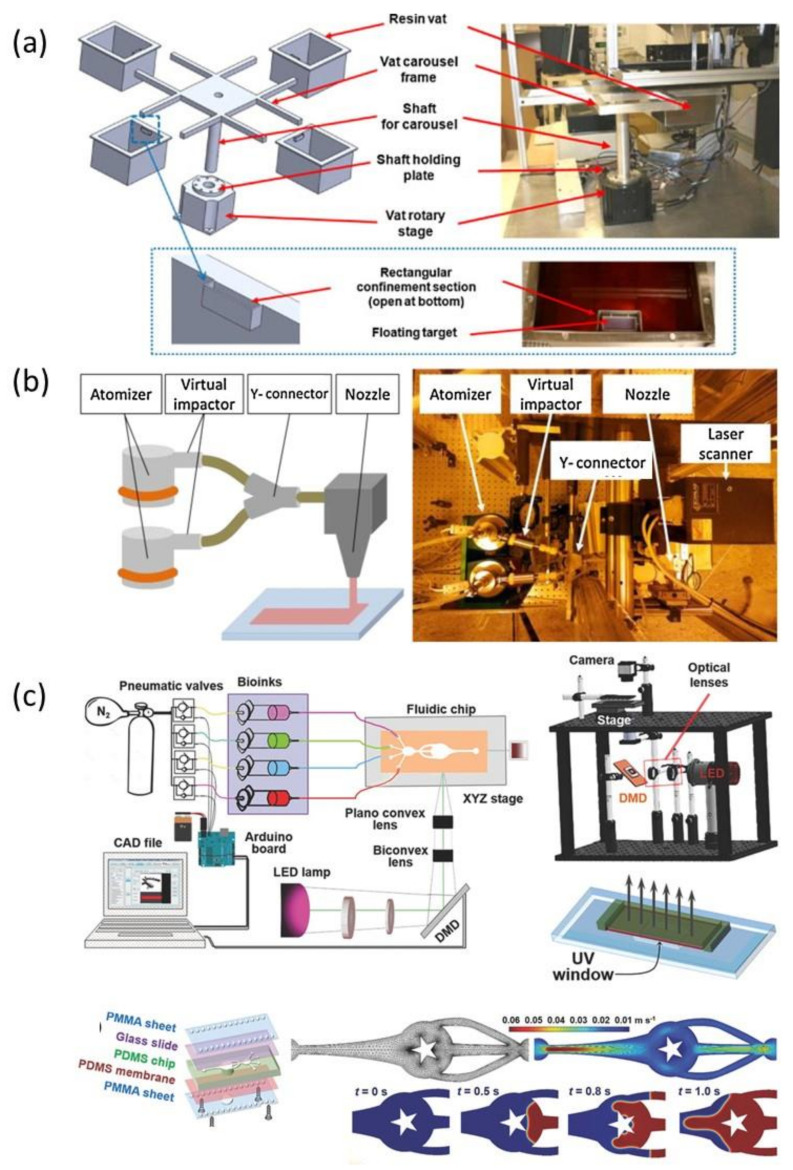
Recent advances in multi-material SLA technique: (**a**) the rotating vat carousel system (images were adapted from Ref. [127]), (**b**) the dual atomizer aerosol jet system (images were adapted from Ref. [128]), and (**c**) the setup of the bioprinter equipped with a UV lamp (385 nm), optical lenses and objectives, a DMD chip, and a microfluidic device (images were adapted from Ref. [129]).

**Figure 11 polymers-13-01957-f011:**
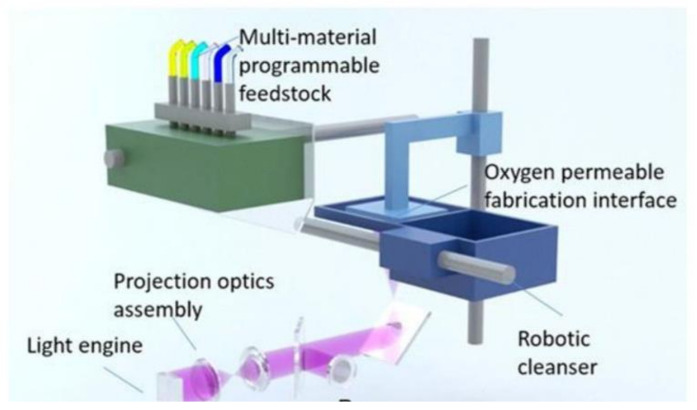
The modular DLP printing technique coupled with in situ-microfluidic systems for resin delivery (images were adapted from Ref. [131]).

**Figure 12 polymers-13-01957-f012:**
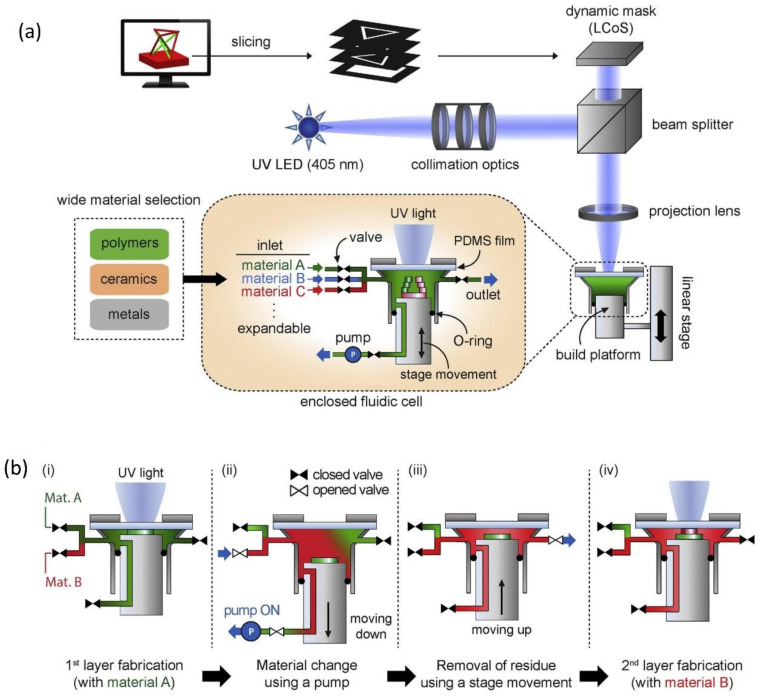
The cleaning process-free multi-material DLP technique combining dynamic fluidic control: (**a**) schematic illustration of this system exhibiting the light source, resin vat, and stage and (**b**) detailing the resin exchange process (i) printing the first layer with material A, (ii) replacing material A with material B, (iii) positioning for the next layer, (iv) curing material B upon UV exposure (images were adapted from Ref. [133]).

**Figure 13 polymers-13-01957-f013:**
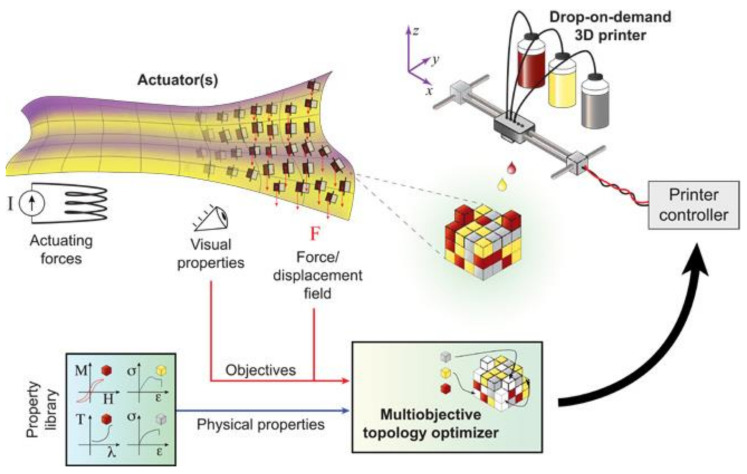
Overview of the specification-driven 3D printing process (images were adapted from Ref. [144]).

**Figure 14 polymers-13-01957-f014:**
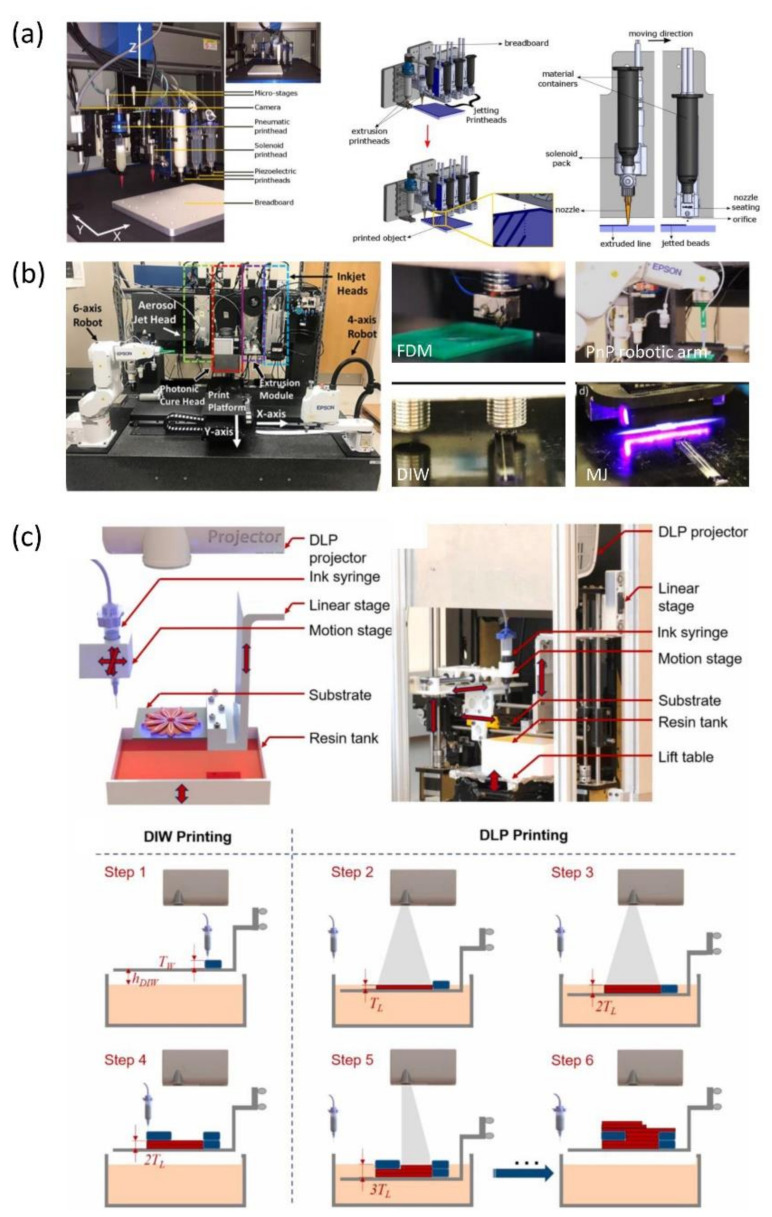
Recent advances in hybrid techniques for multi-material fabrication: (**a**) the hybrid system combining a pneumatic material extrusion print head and a piezo-pneumatic print head (images were adapted from Ref. [154]), (**b**) the m4 hybrid 3D printer integrating inkjet printing, FDM, DIW, aerosol jetting, and a PnP robotic arm (images were adapted from Ref. [155]), and (**c**) the hybrid 3D printing system with integrated DLP and DIW 3D printing techniques (images were adapted from Ref. [156]).

**Figure 15 polymers-13-01957-f015:**
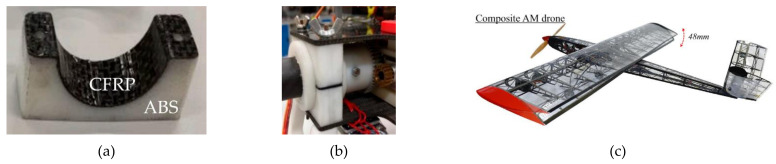
(**a**) CFRP/ABS composite bottom clamp (images were reprinted from Ref. [162]); (**b**) assembled clamp to the joint of the quadcopter (images were reprinted from Ref. [162]); (**c**) additively manufactured composite drone illustrating morphing actuation (images were reprinted from Ref. [164]).

**Table 1 polymers-13-01957-t001:** Top journal outputs from January 2000 to March 2021 published research works related to multi-material additive manufacturing of polymers.

JOURNAL	Number of Relevant Articles	% Total Publications	Number of Total Citations from These Articles	Number of Average Citations per Article
*Additive Manufacturing*	98	5.02%	1828	18.7
*ACS Applied Materials and Interfaces*	92	4.71%	1397	15.2
*Materials*	80	4.10%	913	11.4
*Composites Part B Engineering*	66	3.38%	1951	29.6
*Polymers*	66	3.38%	359	5.4
*Journal of Applied Polymer Science*	65	3.33%	1107	17.0
*Composites Science and Technology*	64	3.28%	1991	31.1
*Materials and Design*	44	2.25%	956	21.7
*Rapid Prototyping Journal*	44	2.25%	934	21.2
*Advanced Functional Materials*	39	2.00%	890	22.8
*Composites Part A Applied Science and Manufacturing*	31	1.59%	732	23.6
*Rsc Advances*	29	1.49%	215	7.4
*Scientific Reports*	29	1.49%	1591	54.9
*Advanced Materials*	24	1.23%	1952	81.3
*Journal of Composite Materials*	24	1.23%	270	11.3
*Advanced Engineering Materials*	23	1.18%	300	13.0
*Journal of Polymer Materials*	23	1.18%	25	1.1
*Materials Science and Engineering C*	23	1.18%	305	13.3
*Polymer Composites*	22	1.13%	171	7.8
*Materials Letters*	21	1.08%	224	10.7
*Sensors and Actuators B Chemical*	21	1.08%	321	15.3
*Composite Structures*	20	1.02%	367	18.4

**Table 2 polymers-13-01957-t002:** List of keywords related to multi-material AM techniques and their relevant network data.

Keyword	Occur.	Links	Total Link Strength	Avg. Pub. Year	Avg. Citations
fused deposition modeling	127	65	167	2019	21.7
stereolithography	35	28	36	2018	18.5
inkjet printing	31	24	30	2016	12.4
selective laser sintering	24	17	22	2018	15.6
material extrusion	21	12	13	2020	3.5
screen printing	19	20	25	2016	18.6
extrusion	17	26	35	2019	15.5
direct ink writing	12	14	14	2020	3.9
photopolymerization	11	5	5	2018	31.8
electrospinning	9	10	11	2016	17.8
digital light processing	7	11	11	2019	24.7
material jetting	6	4	4	2019	35.7
powder bed fusion	5	3	4	2020	2.6

**Table 3 polymers-13-01957-t003:** List of keywords related to multi-material AM applications and their relevant network data.

Keyword	Occur.	Links	Total Link Strength	Avg. Pub. Year	Avg. Citations
tissue engineering	35	24	51	2018	38.6
scaffold	28	23	46	2016	51.8
printed electronics	24	22	34	2017	26.6
bone tissue engineering	16	10	14	2019	15.1
sensor	13	14	17	2018	16.5
bone regeneration	11	11	13	2019	14.5
flexible electronics	11	11	13	2018	37.0
strain sensor	10	10	14	2017	19.0
electrochemical sensor	7	2	2	2018	13.6
drug delivery	6	3	3	2019	13.2
temperature sensor	6	8	9	2018	14.0
humidity sensor	6	3	3	2018	9.0
dielectrics	6	6	7	2017	3.8
biosensor	6	3	3	2017	12.3
pressure sensor	6	7	7	2015	12.3
embedded capacitor	6	4	7	2006	58.7
actuators	5	8	9	2019	20.6
soft robotics	5	6	7	2019	17.6
wearable electronics	5	3	3	2019	6.0
stretchable electronics	5	4	5	2018	17.2

**Table 4 polymers-13-01957-t004:** List of authors that published the most publications related to multi-material additive manufacturing of polymers.

Scholar	Num. of Publications	Citations	Avg. Pub. Year	Avg. Citations
Wang Y.	32	928	2018	29.0
Wang J.	25	503	2017	20.1
Liu Y.	23	335	2018	14.6
Zhang J.	21	206	2019	9.8
Zhang X.	21	269	2018	12.8
Li Y.	20	201	2019	10.1
Singh R.	20	287	2019	14.4
Zhang Y.	20	367	2018	18.4
Wang L.	18	476	2017	26.4
Zhang H.	17	125	2018	7.4
Zhang Z.	17	275	2019	16.2
Wang Z.	16	583	2018	36.4
Zhang C.	16	307	2018	19.2
Chen Y.	15	514	2018	34.3
Xu J.	15	220	2019	14.7

**Table 5 polymers-13-01957-t005:** List of countries/regions published the most publications related to multi-material additive manufacturing of polymers.

Country/Region	Num. of Publications	Citations	Avg. Pub. Year	Avg. Citations
United States	519	17544	2017	33.8
China	378	6450	2018	17.1
United Kingdom	147	3541	2017	24.1
Germany	145	1925	2017	13.3
South Korea	139	2801	2016	20.2
India	112	1318	2017	11.8
Italy	91	2334	2017	25.6
Spain	71	1269	2018	17.9
France	66	1172	2018	17.8
Australia	62	1499	2018	24.2
Canada	60	477	2018	8.0
Taiwan	55	944	2016	17.2
Russian Federation	52	668	2018	12.8
Singapore	50	2323	2017	46.5
Japan	45	2492	2014	55.4
Switzerland	44	1220	2017	27.7
Poland	39	307	2018	7.9
Iran	34	270	2019	7.9
Sweden	34	452	2017	13.3

**Table 6 polymers-13-01957-t006:** Composite ceramics and polymers scaffolds fabricated using AM for tissue engineering [189].

Ceramics	Polymer	AM Technology	Reference
Calcium chloride, glutamic acid, ammonium hydrogen phosphate	Sodium alginate	Material extrusion	[190]
Titanium, platelets	Gelatin	PBF	[191]
HA, solvent system	PLGA	Material extrusion	[192]
Calcium silicate, magnesium PCL Laser sintering	PCL	PBF	[193]
HA, PLGA microspheres	PCL	Material extrusion	[194]
Graphene	PCL	Material extrusion	[195]
HA	PCL	Material extrusion	[196]
BCP	PLGA, PCL, collagen	Material extrusion	[197]

## Data Availability

The data presented in this study are available on request from the corresponding author.

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
