# Peer review of "Scientometric Analysis and Systematic Review of Multi-Material Additive Manufacturing of Polymers"

_polymers, 2021, doi:10.3390/polym13121957_

Round 1

Reviewer 1 Report

The following proposed edits are optional:

The Abstract and Conclusions could be improved by the inclusion of the scientific/technological findings of the conducted review.

Reviewer 2 Report

The paper Scientometric analysis and Systematic review of multi-material
additive manufacturing of polymers is based on Multi-material additive-manufacturing of polymers. The main steps proposed contribute to providing knowledge for practitioners and researchers to understand the state of the art on these materials. 2512 papers were analyzed using the “scientific mapping” approach and a systematic review from scopus database on the latest advances in multi-material AM of polymers.

This issue is not a novelty in this area since it is a state-of-the-art paper. Although, I agree with the authors that it can provide knowledge collection to practitioners and researchers.

The paper would benefit from a better and more detailed discussion, highlighting the novelty of the work and defining research alternatives paths to the identified needs.And how to explore the potentials found, how to develop and improve post-processing methods. I recommend reconsideration of the paper following major revision.

Abstract

Trendare-please correct it

Round 2

Reviewer 2 Report

Answers to the comments and suggestions were performed. We suggest accepting in the present form.